# The impact of mental health literacy intervention on in-service teachers' knowledge attitude and self-efficacy

Azziz Bichoualne[1] ◉, Mohamed Oubibi[2] ◉ and Yu Rong[1]

[1]College of Education. Zhejiang Normal University, Jinhua, China and [2]Smart Learning Institute of Beijing Normal University, College of Education, Beijing, China

mental health knowledge; stigma; mental health; mental health literacy training; self-efficacy

**Corresponding author:**
Azziz Bichoualne;
Email: azziz.bichoualne@zjnu.edu.cn

## Abstract

Mental health literacy (MHL) is an essential component in the process of de-stigmatization, promoting mental health, and supporting people struggling with mental illness. Today, as the number of people suffering from mental illnesses is nearly 450 million people worldwide, the importance of having a teacher-training program that incorporates MHL in its curriculum has become paramount. This study is a quasi-experimental pre–post research that uses the MHL program as an intervention. The sample included 36 ($n = 36$) high school in-service teachers from 11 schools. The participants took an online training program for 6 weeks. The evaluations were used before and after the training to assess their mental health knowledge, attitude, and self-efficacy. The data were analyzed using the paired sample *t*-test. The findings revealed a significant level of improvement in relation to the teachers' MHL knowledge, attitude, and self-efficacy after the training. The study findings reveal the effectiveness of MHL training in improving in-service teachers' knowledge, attitude, and self-efficacy in relation to mental health. The study's limitations and future study recommendations are discussed.

## Impact statement

The results of this study demonstrate the profound impact of incorporating mental health literacy (MHL) into teacher training programs. The quasi-experimental pre–post research, involving 36 high school in-service teachers, unequivocally illustrates the effectiveness of a 6-week MHL program intervention. This research provides compelling evidence for the efficacy of MHL training in equipping in-service teachers with the knowledge, attitudes, and self-efficacy necessary to address mental health concerns effectively.

## Introduction

According to the World Health Organization (WHO, 2022), mental health problems are one of the leading behavioral adjustment difficulties in children. Its implications often result in the inability to perform functions, adapt to challenges, and establish healthy relationships with others. Studies report that there is nearly 15% of young people between the ages of 10 years and 19 years suffering from a mental health disorder in the world (WHO, 2021), 10% of which receive no adequate intervention at an early age (The Children's Society, 2019). While these rates are high, they stand distinguishably higher in low-income countries (Collins et al., 2011). North African countries were reported to have close numbers, with 25% of adolescents suffering from anxiety disorders in Egypt and 18.7% suffering from anxiety and depression in Tunisia (Mzadi et al., 2022). Among the factors that exacerbate the mental health situation in African countries is the negative attitude toward mental health that often takes the form of a poor understanding of mental health and stigmatization (Crabb et al., 2012). Due to the scarcity of research on the attitude toward mental health, several views have suggested that African countries are less impacted by stigma (Sewilam et al., 2014; however, recent studies discredited the perception and have shown the presence of stigma and its impact in various African countries such as Morocco and Ethiopia (Greene et al., 2021).

While most of the literature on mental health is derived from research conducted in high-income nations, where there is higher awareness and regular screening of patients with mental problems, in contrast, dealing with mental problems in low- and middle-income countries is neglected, under-resourced, and under-funded. Consequently, knowledge of the effects of mental problems on people from low- and middle-income countries is still lacking (Alloh et al., 2018). Moreover, poor access to mental health care is increasing, with several nations reporting rates of less than 50% to less than 10% of people who have access to mental care (Patel et al., 2010).

Morocco is a lower-middle-income country located in North Africa with a population of 37 million people (United Nations Department of Economic and Social Affairs, 2022). In 1920, it had its first psychiatric institution, followed by the creation of a dozen others, along with establishing legislative measures that offered an operational framework for mental institutions in the following years (WHO, 2006). According to the Economic, Social and Environmental Council (2022) in Morocco, in a national study report completed by the High Commission for Planning, 49.4% of the respondents stated that they suffered from anxiety, while 40.9% suffered from fear (29.6% from claustrophobia and 24.4% from multiple phobias), 23.7% from sleep disturbance, 7.5% from hypersensitivity/nervousness, and 5.9% of depression.

Health literacy is a more effective element in predicting health than education, income, race, or professional status (Joseph et al., 2019). MHL was first used by Jorm et al. (1997) as a concept that refers to the social understanding of mental health and the ways to manage it (Furnham and Swami, 2018). When MHL was first introduced, it had an impact on multiple policies that influenced ways of intervention (Lo et al., 2017). The concept of MHL was expanded afterward to cover the understanding of positive mental health, mental illnesses, treatment, stigma, and the promotion of help-seeking behavior and ways to help others (Kutcher et al., 2015). Further, the concept evolved to cover specific mental health disorders such as ADHD, schizophrenia, and many others (Furnham and Blythe, 2012). A conducted study reported that half of the mentally ill cases do not have any treatments for their conditions in developed countries (Jansen et al., 2015), and the lower levels of health literacy were related to the high levels of inequality. These situations may explain the relationship between lack of improvement in mental health conditions and failure in early help-seeking and early intervention (Henderson et al., 2013).

In the school environment, low levels of MHL among educators are associated with inadequacy in recognizing and managing students with mental health concerns (Singh et al., 2022). As students spend much of their time in school, the latter plays an important role in helping them manage their mental health (Abdinasir, 2019). Schools are a gateway to offering help to students with mental health challenges and increasing their accessibility to mental health services (Thorley, 2016). Since the role of teachers has been confined to almost excluding the provision of mental health aid to students, assisting teachers to help students with mental health problems has also been narrowed down (O'Reilly et al., 2018). With adequate support, teachers could play an essential role in identifying students' mental health needs and helping them with ways to get help, which, in turn, can help stop the decline of students' mental health conditions (Grove et al., 2015).

The teacher mental health literacy (TMHL) school-based program was created by Kutcher et al. (2013) to provide educators with information in a framework of steps in order to guide them in identifying and helping students with mental health disorders (Anderson et al., 2019). According to Miller et al. (2018), the TMHL program positively improves teachers' knowledge and attitude toward students with mental health disorders; in addition, it boosts the teachers' sense of confidence in implementing new practices. Moreover, Kutcher et al. (2016) suggested that the program could help the participants by 1) expanding positive mental health knowledge, 2) narrowing down stigma, and 3) promoting the efficacy of finding help. According to Kutcher et al. (2014), the program is based on evidence to help teachers broaden their understanding of mental health. The models utilized provide both a framework of

attitude and guidance to people looking for treatment for their conditions (Demyttenaere et al., 2004). Lo et al. (2017) argue that this expansion has led to several interventions in different educational policies in different countries.

Mental health literacy (MHL) was significantly and substantially improved by providing teachers with knowledge and resources on mental health and assisting them in incorporating it into their already existing professional abilities through the TMHL program. In addition, different studies' findings reported that the training significantly decreased teachers' mental health stigma, improved their attitudes toward mental problems significantly, and increased their understanding of help-seeking strategies following the program in different countries, such as Malawi (Kutcher et al., 2015), Canada (Kutcher and Wei, 2014; Kutcher et al., 2014), and Tanzania (Kutcher et al., 2016).

Self-efficacy is defined as a "belief in one's capabilities to organize and execute the courses of action required to manage prospective situations" (Bandura, 1995, p. 2). It is one of the major mechanisms in human beings to provide the ability to manage independent thinking and have control over their lives (Bandura, 2001). There are other factors that serve as motivators and guides, but a person believing in their capabilities significantly impacts their thinking, feelings, and actions (Cook and Artino, 2016).

Heng and Chu (2023) highlight that studies demonstrate a strong association between teachers' sense of self-efficacy and their ability to identify students with mental health issues. Self-efficacy in teachers, which is enhanced by appropriate training, is pivotal for supporting students affected by anxiety, depression, and other mental health challenges, and for implementing early intervention strategies. Furthermore, Giallo and Little (2003) have found that effective classroom management correlates with teachers' confidence in their capabilities, a key component of teacher self-efficacy. Their research underscores the connection between the belief in one's actions leading to expected outcomes. In essence, classroom management is intertwined with self-efficacy, readiness, and experience: teachers with greater self-efficacy tend to feel more prepared and exhibit better classroom management.

## Method

This study is quasi-experimental research implemented to investigate the relationship between high school teachers' program training and their knowledge, attitude, and self-efficacy toward mental health. This research is an empirical interventional study used to estimate the causal impact through pre–post intervention applied to the sample population. Thirty-six ($n = 36$) high school teachers from 11 schools located in three urban areas have participated in this study. The sampling in this research is carried out in a purposeful way. The participants in this study were in-service high school teachers working within the public Moroccan educational system. In addition, the participants' teaching, ranged from different teaching subjects (language subjects, science subjects, and physical education subjects). The participants consisted of 21 male and 15 female participants. Thirty-three participants are between the age of 25 years and 50 years, followed by two participants under the age of 25 and one participant over 50 years old. Thirty-four participants have a bachelor's degree, followed by one participant with a master's degree, and one with a Ph.D. degree. Regarding the participants' years of experience, twenty-six participants' experience ranges between 1 year and 5 years, followed by seven

**Table 1.** Demographic profile

| Variable | Item | Frequency | % |
|---|---|---|---|
| Gender | Male | 21 | 58.3 |
| | Female | 15 | 41.6 |
| Age | ≤25 | 2 | 5.5 |
| | 25–50 | 33 | 91.6 |
| | >50 | 1 | 2.7 |
| Educational background | Bachelor | 34 | 94.4 |
| | Master | 1 | 2.7 |
| | Ph.D. | 1 | 2.7 |
| Years of experience | (1–5 years) | 26 | 72.2 |
| | (5–10 years) | 7 | 19.4 |
| | ≥10 years | 3 | 8.3 |
| Teaching subject | English | 12 | 33.3 |
| | French | 8 | 22.2 |
| | Physics and chemistry | 2 | 5.5 |
| | Biology and geology | 6 | 16.6 |
| | Mathematics | 5 | 13.8 |
| | Philosophy | 2 | 5.5 |
| | Physical education | 1 | 2.7 |

participants with an experience between 5 years and 10 years, and three participants with over 10 years of experience. The participants' characteristics are shown in Table 1.

TMHL training program is used as an intervention in this research. The training program' curriculum, which has been put openly for usage (Kutcher et al., 2013), consists of six modules: the stigma of mental illness; understanding mental illness and wellness; information about specific mental illnesses; experiences of mental illness; seeking help and finding support; and the importance of positive mental health. Each of these modules provides other material that helps learners to better their understanding of mental health through extensive resources, lesson plans, classroom activities, lesson summaries, and learning objectives. The phase of pre-testing in a quasi-experiment research provides a measurement of the characteristics of the participants before the treatment. In contrast, post-testing provides a measurement of the characteristics of the participants after receiving the treatment. The TMHL program has been developed to help enhance the MHL of the participant teachers through four steps: 1) understanding how to optimize and maintain good mental health, 2) understanding mental disorders and their treatments, 3) decreasing stigma, and 4) enhancing help-seeking efficacy. The six modules are intended to be taught sequentially. In each module, there are two components: core materials and supplementary materials. The core materials are intended for use in the classroom to attain the objectives of the program. The supplementary materials are intended for participants who desire to devote extra time and effort to learning about the module topic. Each module has the following components: 1) an overview that offers a summary of the module; 2) learning objectives that outline particular understandings or abilities that participants will gain from completing the modules; 3) major concepts section presenting the major themes addressed within the module;

4) activities section, which contains information about suggested classroom applications; 5) required materials section, which contains resources needed to complete the activities in each module; and 6) in advance section, which contains instructions for gathering and preparing materials needed to complete the activities in the module.

The training intervention was implemented online over a 6-week period. Each week, participants were given a 2-h course from the TMHL curriculum modules. Prior to the initiation of the program, participants were informed through their email of the schedule of the date and duration of each course session. Additionally, participants were provided with a form of consent that contained a full explanation of the program and the anonymity of their participation.

The first session was introductory to explain to the participants the training courses and the scheduling of lessons during the six following weeks. Prior to the beginning of the first course, the participants were handed three surveys: 1) a 30-item pre-test survey of the TMHL curriculum guide program; 2) the Devaluation of Consumer Scale (DCS); and 3) the Student Mental Health Self-Efficacy Teacher Survey (SMH-SETS).

The participants completed the surveys before the first session of the course began. Every module was designed to take a 2-h time during the weekend of every week. In addition, the participants were given a quiz at the end of every module to ensure they had grasped the lessons and were provided with the material used during the class. The participants were given extensive reading resources after every session to help them further and guide their understanding of the content discussed in each module.

During the sixth week of the course, the participants were given the post-test survey of the TMHL curriculum guide program, the DCS post-survey, and the SMH-SETS post-survey to measure the impact of the training on their level of knowledge, attitude, and self-efficacy toward mental health illness. The knowledge test was evaluated through 30 questions with responses "true," "false," or "I do not know" responses.

The DCS was first created to measure the extent of the caregiver's belief in the devaluation of people with mental illness, and it consists of three factors, namely status reduction, role restriction, and friendship refusal (Chang et al., 2018). The DCS applied in this study is based on the version that was tested and proven to be valid and reliable, with an overall scale reliability of 0.71 by Struening et al. (2001).

Through DCS, the attitude toward mental health illness was measured before the intervention of the MHL curriculum guide program and right after it to test the impact of the intervention on the attitude of participants. The DCS uses a scale of eight items on a seven-point Likert from 1, "strongly agreeing," to 7, "strongly disagreeing." The statements phrased in the scale using negative structures, for example, "Most people who have a mental illness are dangerous and violent" (item 1); disagreeing with the statements means having a positive attitude toward it.

According to Brann et al. (2020), the SMH-SETS is a measurement used to see the belief of competence in people's professions. The SMH-SETS do so by showing professional competence in recognizing students' mental health status, dealing with students' mental needs and concerns, and encouraging the mental health subject to be addressed in school environments (Brann et al., 2020). The SMH-SETS applied in this study was based on the instrument designed by Brann et al. (2020). It has 15 questions that are scaled on a six-point Likert scale ranging from 1 (strongly disagree) to 6 (strongly agree). The questions in the SMH-SETS begin with the

**Table 2.** Internal consistency

| Scale | Cronbach's alpha |
|---|---|
| Pre-intervention mental health knowledge | 0.741 |
| Post-intervention mental health knowledge | 0.783 |
| Pre-intervention mental health attitude | 0.743 |
| Post-intervention mental health attitude | 0.782 |
| Pre-intervention mental health self-efficacy | 0.834 |
| Post-intervention mental health self-efficacy | 0.937 |

statement "I feel confident in my ability to…." Sample questions include "I feel confident in my ability to respond when a student is displaying aggressive behavior." the maximum summed score for the SMH-SETS is 90 while the minimum score is 15, with excellent internal consistency (Cronbach's alpha = 0.91), item reliability and validity (Brann et al., 2020).

The statistical analysis was carried out using the software IBM® SPSS® (version 27). The paired sample *t*-test was used for the data gathered from 36 participants (*n* = 36) who completed the pre-test and the post-test to compare and assess the change in the scores of their mental health knowledge, attitude, and self-efficacy.

## Results

### Participants' characteristics results

Thirty-six people (*n* = 36) participated in this study, with 21 males (*n* = 21, 58.3%), and 15 females (*n* = 15, 41.6%). Thirty-three (*n* = 33, 91.6%) participants age is between 25 years and 50 years, followed by two participants (*n* = 2, 5.5%) under the age of 25 and one participant (*n* = 1, 2.7%) over 50 years old. Regarding the educational background of the participants, thirty-four participants (*n* = 34, 94.4%) have a bachelor's degree, followed by one participant (*n* = 1, 2.7%) with a master's degree, and one participant (*n* = 1, 2.7%) with a Ph.D. degree. The distribution of the participants' subject of teaching range as follows: 1) 20 participants in languages (*n* = 20, 55.5%) divided between twelve English teachers (*n* = 12, 33.3%) and eight French language teachers (*n* = 8, 22.2%); 2) 13 teachers of sciences (*n* = 13, 36.1%) divided between two physics and chemistry teachers (*n* = 2, 5.5%), six biology and geology teachers (*n* = 6, 16.6%), and five teachers of mathematics (*n* = 5, 13.8%); 3) two teachers of philosophy (*n* = 2, 5.5%); 4) one teacher of physical education (*n* = 1, 2.7%). In relation to the years of experience, twenty-six (*n* = 26, 72.2%) participants' experience ranges between 1 year and 5 years, followed by seven participants (*n* = 7, 19.4%) with an experience between 5 years and 10 years, and three participants (*n* = 3, 8.3%) with over 10 years of experience. The results are shown in Table 1.

### Mental health knowledge results

The internal consistency results of TMHL in this study were acceptable: Cronbach's *α* pre-test = 0.741; Cronbach's *α* post-test = 0.783. The results are shown in Table 2. The mental health knowledge score of the pre-test and post-test ranged from 7 to 29, with 30 being the highest score possible. The paired sample *t*-test was conducted to evaluate the difference in scores between the pre-test and post-test. As shown in Table 3, the results indicate that there was a statistically significant increase in the mental health knowledge between the score of the pre-test results (*M* = 14.77, SD ± 4.26) and the post-test results (*M* = 18.88, SD ± 4.502) with (*t* = 5.366, *p* < 0.001); Cohen's *d*: 0.894. The effect size in this study is in accordance with Cohen's guidelines for a large effect (*d* ≥ 0.8).

### Teachers' attitude toward mental health results

The internal consistency was measured and indicated acceptable results with Cronbach's *α* pre-test = 0.743; Cronbach's *α* post-test = 0.782. The results are shown in Table 2. The teachers' attitude scores range from 15 to 39, with 56 being the highest score possible. About 60% of the participants' responses represented with "disagree," and "somewhat disagree," while the question "I would be willing to have a person with a mental illness at my school." had the highest negative responses. As shown in Table 3, the results from the paired sample *t*-test indicate a statistical significance in the attitude from the evaluation scores of pre-intervention (*M* = 20.72 SD ± 5.401) and the post-intervention (*M* = 26.47, SD ± 5.852) with (*t* = 5.900, *p* < 0.001); Cohen's *d* = 0.983. The effect size in this study is in accordance with Cohen's guidelines for a large effect (*d* ≥ 0.8).

### Self-efficacy results

The internal consistency was measured and indicated acceptable results with Cronbach's *α* pre-test = 0.834; and Cronbach's *α* post-test = 0.937. The results are shown in Table 2. The scores of teachers' self-efficacy ranged between 33 and 58 with 90 being the highest possible score. The results from the paired sample *t*-test indicate a statistical significance in the self-efficacy from the evaluation scores of pre-intervention (*M* = 41.00, SD ± 7.10) and the post-intervention (*M* = 48.80, SD ± 7.07) with (*t* = 4.371, *p* < 0.001); Cohen's *d* = 0.729. The effect size in this study is in accordance with Cohen's guidelines for a medium effect (*d* = 0.50). The results are shown in Table 3.

## Discussion

The significance of MHL in combating stigma, promoting mental well-being, and providing support for individuals struggling with mental health issues cannot be overstated (Wang et al., 2020). With approximately 450 million people worldwide grappling with mental illnesses, the integration of MHL into teacher training programs

**Table 3.** Pre–post scores of mental health knowledge, attitude, and self-efficacy

| Instrumentation | Pre-intervention mean | Pre-intervention SD | Post-intervention mean | Post-intervention SD | Cohen's *d* | *t* | *P*-value |
|---|---|---|---|---|---|---|---|
| Mental health knowledge | 14.77 | 4.26 | 18.88 | 4.502 | 0.894 | 5.366 | *p* < 0.001 |
| Mental health attitude | 20.72 | 5.401 | 26.47 | 5.852 | 0.983 | 5.900 | *p* < 0.001 |
| Mental health self-efficacy | 41.00 | 7.10 | 48.80 | 7.07 | 0.729 | 4.371 | *p* < 0.001 |

has become imperative (Yang et al., 2019). The study focusing on the impact of a MHL intervention on high school in-service teachers' knowledge, attitudes, and self-efficacy presents significant implications for both education and mental health support within educational settings.

Having the TMHL program implemented in this study for 6 weeks has shown to prove with statistical significance the participants' outcome of mental health knowledge, attitude, and self-efficacy toward mental health illnesses. The originality of this research lies in being among the few studies conducted on MHL within educational settings in North Africa in general and Morocco in specific. The results show a resemblance of impact with other studies done on TMHL in educational settings in Canada, Malawi, and Tanzania (Kutcher et al., 2016). As often teachers' roles are associated with simply giving knowledge and content along with managing students (Mahlios, 2002), this study highlights the importance of teachers' roles and expands upon its significance as an essential construct to recognize and support students struggling with mental health illnesses as well as help them find assessment and treatment. Many studies report that the need for intervening early, identifying the problem, and treating children and youth is important to alleviate the symptoms, achieve higher rates of recovery, and improve results among children and youth in general (Trudgen and Lawn, 2011). The importance of this study comes as a part of students' early intervention, as it accentuates the importance of providing teachers with adequate knowledge to recognize, support, and guide students with mental health illnesses.

The study revealed a significant and positive transformation in the participants' MHL knowledge, attitude, and self-efficacy subsequent to the training. This underscores the effectiveness of MHL training in empowering in-service teachers with enhanced understanding, more empathetic attitudes, and greater confidence in dealing with mental health challenges. Such outcomes hold tremendous promise for fostering more supportive and inclusive educational environments, ultimately benefiting the mental well-being of students. Despite the improvement in mental health knowledge attitude and self-efficacy among participants in this study, it also shows limitations to consider. This study has a small sample size, which may cause a limitation of the validity of the research. Hence, it is advised to implement a larger size to improve the accuracy and reliability in future studies. Unlike other studies conducted with TMHL in Africa, this study used the French language-based version of the TMHL; however, it was not contextualized to match the educational Moroccan environment. Therefore, additional research is needed in this area for the purpose of having more understanding of the effectiveness and long-term impact of the MHL on the knowledge, attitude, and self-efficacy of teachers toward mental health illnesses. The implementation of this study among high school teachers is another limitation of the study. Future studies are recommended to extend the scope of their samples to include other school levels.

This research provides compelling evidence for the efficacy of MHL training in equipping in-service teachers with the knowledge, attitudes, and self-efficacy necessary to address mental health concerns effectively. By integrating MHL into teacher education, educational institutions can play a pivotal role in destigmatizing mental health issues and creating a more supportive environment for all students. This study stands as a significant step forward in the broader effort to prioritize mental health within educational settings.

## Conclusion

This study aimed to assess the impact of MHL training on the teachers' mental health knowledge, attitude, and self-efficacy. The findings of the study revealed a statistically significant improvement among the participants. Correspondingly, studies on MHL conducted in African countries reported similar results (Kutcher et al., 2016). This study calls for initiatives for emerging similar training in both pre-service and in-service teacher training to help mitigate the negative influence of stigmatization and misconception in both students and teachers. Further, this study builds on the perspective that challenges the boundaries set for the role of teachers as only conveyors of information and tries to help regain the discourse of integrating schoolteachers in the process of mental health intervention through professional training.

**Open peer review.** To view the open peer review materials for this article, please visit http://doi.org/10.1017/gmh.2023.77.

**Data availability statement.** The data that support the findings of this study are available on request from the corresponding author, A.B. The data are not publicly available due to restrictions as they are containing information that could compromise the privacy of research participants.

**Acknowledgments.** We express our gratitude to Ms. Melissa Smith for her tremendous guidance, support, and encouragement throughout this study. We would like to extend our heartfelt thanks to all the participants in this study, who generously shared their time, experiences, and insights. Their willingness to engage in this research was essential to the success of this study, and we are deeply grateful for their participation.

**Author contribution.** The authors confirm sole responsibility for the following: study conception and design, data collection, analysis and interpretation of results, and manuscript preparation.

**Financial support.** This research received no specific grant from any funding agency, commercial, or not-for-profit sectors.

**Competing interest.** The authors have no competing interests to declare.

**Ethics statement.** The author asserts that all procedures contributing to this work comply with the ethical standards of the relevant national and institutional committees on human experimentation and with the Helsinki Declaration of 1975, as revised in 2008.

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
