## [Reviewer Report]

Azziz Bichoualne

Zhejiang Normal University

Zhejiang Normal University. No.688, Yingbin Road, Jinhua, Zhejiang Province, China.

azziz.bichoualne@zjnu.edu.cn, +8617858990701

Victoria Lane

Cambridge Prisms: Global Mental Health

06/21/2023

Dear Editor,

Please consider my research paper titled “ Mental Health Literacy Intervention: The Impact on In-Service Teachers’ Knowledge Attitude and Self-Efficacy.” for publication in Cambridge Prisms: Global Mental Health. 

The manuscript describes the incorporation of mental health literacy programs in the training of high school educators. The results of the study revealed a significant level of improvement in relation to the teachers’ mental health literacy knowledge, attitude, and self-efficacy after the training. 

I believe that these findings will be interesting to the readers of Cambridge Prisms: Global Mental Health as they are in accordance with the broad application of ‘the global point of view’ of mental health issues.

The preparation of the program, the translation of material, and the application of the mental health literacy program have been very challenging, yet successful tasks that led to the production of this study. 

This manuscript has not been published and is not under consideration for publication elsewhere. 

Thank you for considering this manuscript!

Azziz Bichoualne, PhD

---

## [Reviewer Report]

Please check the introductory text to ensure researcher’s analytical or critical voice on the study. You may identify or throw more light on the theoretical underpinning of your research. You are also encouraged to utilize proper academic writing tone as some of your sentences appear loosely connected and a bit confusing.

introduction, pg 2, ln 11 -change percents to percent

p2, ln 26 - 18,7 percent. Do the authors mean 18.7?

pg 3, ln 6 - change when to When

pg 3, ln9-15 “The concept of MHL was expanded afterward to cover positive mental health understanding, mental health illnesses understanding andtheir treatment, understanding the impact of stigma on mental health, and promoting help-seeking strategies and treatment options and ways to help others.” You could consider the following statement as a replacement-To cover the understanding of positive mental health, mental illnesses, treatment, stigma and the promotion of help seeking behavior and ways to help others.

pg 3, ln 30-37 - What is the evidence that the MHL training has worked and is effective in some climes? 

pg3, ln 46 - change help to helping

pg 3, ln 48 - change condition to conditions

pg 3, ln 51-53 - Please clarify this statement as it appears conflicting

pg 4, ln 6 - change it to them

pg 4, ln 15 - “narrowing down the increase of stigma” change to narrowing down stigma

pg 4, ln 18 - “evidence to be helping teachers ” change to evidence to help teachers

pg 4 ln37 change feeling to feelings

pg 5, ln 7-23 - You may consider looking at a particular theory. Adopting this theory entails explaining the rationale for your choice of the theory. Explain why it is pertinent and practicable within the confines of your study.

pg 5 ln 32 - “mental health illness” - Please check and identify the best language to use here ‘either mental illness or mental health

pg 5, ln 37 - “without randomisation” - Delete if you may. A quasi experiment implies no randomization

pg 6, ln 2-4 “the participants that partook in the process of providing input to the study were considered in advance and chosen.” - Needless to state. I suggest you delete

pg 6, ln 23 - You could replace mishaps with –‘unfavorable outcomes’as mishap appears too strong here

pg 6, ln 40 - delete filling out

pg 6, ln 40 - change begins to began

pg 7, ln 53 - delete have

pg 7, ln 53-58 - Please provide teacher’s age range if possible

---

## [Reviewer Report]

Dear Editor,

Apologies for slight delay in completing this review.

Comments below are in good faith and meant to improve the quality of your paper.

Title:

The title reads:

Mental Health Literacy Intervention: The impact on in-service teachers’ knowledge, attitude, and self-efficacy.

Please review grammar.

Should this be … The impact of in-service teacher’s knowledge, attitudes, and self-efficacy.

Abstract:

(Page 2)The first two opening sentences are grammar-error prone and need complete review and rewordingConsidering the abstract is what leads one to proceed reading or move on from your journal article, concise and grammatically correct sentences are important.

I take issue with:

Line7-8: Today as the number of suffering from mental illness is increasing… > Could be clearer, use brief global statistics instead

Line 10: ..resources for promoting MHL are getting shorter > What does this mean?

Key words:

Are key words available for this article?Consider including key words

Introduction:

Review and update Introduction section of the paper. Analysis relies on dated literature that is 20-30 years old.

This section presents some description of mental morbidity for Morocco and similar low and middle-income countries. Unfortunately, the presentation is mostly descriptive and lacking in critical analysis. While I am aware of paucity of published literature on mental health/literacy from North Africa and similar countries, I note most of the references used to be 15 years or older.

Please revise and resubmit a well analysed introduction, that includes current literature on this topic.

Methods:

Methodology is mostly sound, but further clarification is required.Authors have presented their methodology casually, and as a reader, I am unclear how this study can be reproduced elsewhere. Authors must clarify the following:

•Describe the pre-and post-testing instruments (SMH-SETS),and their properties, were these instruments designed for this study or adapted from elsewhere? Is SMH-SETS validated?

•Describe fully the TMHL program and its content

•Demographic characteristics of participants are not included

•Study setting is unclear, is it high schools or teacher training college/university?

•Were participants based in rural/urban/remote setting. These are important considerations when undertaking health literacy research.

•Authors state on Page 6, Line 21-23: ‘Further, participants were informed of the software (ZOOM) that will be used for the delivery of the courses to avoid mishaps’ What does this mean? Statement is vague and does not tell the reader what the authors were controlling for by using Zoom in the study.

Results:

Results Table -Consider including a result table with some output data from IBM SPSS in the body of the paper and not as appendices.

Discussion:

Discussion is rather brief and not well supported with current literature.

Authors are missing an opportunity to use findings from their study, and to present deeper analysis on this topic. From my perspective, I would rather have a shorter Introduction section, but more substantial Discussion section.

Overall Comment:

Authors should be congratulated for this research whose outcomes will contribute to further understanding of mental health literacy among teachers in Morocco and similar low and middle-income countries.

Above review is in good faith for authors to consider and enhance their paper.

---

## [Reviewer Report]

Please respond to the comments by the reviewers. From my end

1. The introduction is too long and needs to be cut by almost 50%.

2. The discussion section is too brief as pointed by one reviewer and needs to be expanded.

3. it is quite natural to see a change in score following an intervention. What are the long term implications of such? I believe the value of this research has this as a major limitation and that needs to be clearly articulated. 

4. what does this add to the knowledge gap in that specific region and what are the implications of it.

---

## [Reviewer Report]

Thank you for the opportunity to review this paper.

I have noted that reviewer suggestion relating to grammar and punctuation, and clearer explanations of methodology have been updated. I also note a frequency table on participant characteristics has been included - that is helpful to readers.

Thank you again for your efforts in exploring this important mental health literacy dimension as it relates to teachers in Morocco.

Best wishes!

Elijah

---

## [Reviewer Report]

Good work. please state the theoretical underpinning of your work. Also make sharp your problem statement in order to tie it to the solutions being brought to bare by your study.

Check and correct a few typos. Congratulations and all the best in your interventions.